# Numerical Study of Bond Slip between Section Steel and Recycled Aggregate Concrete with Full Replacement Ratio

**Chao Liu [1,2,\*], Lu Xing [1], Huawei Liu [1], Zonggang Quan [2,3], Guangming Fu [2,3], Jian Wu [4], Zhenyuan Lv [1] and Chao Zhu [1,2]**

[1]   College of Science, Xi'an University of Architecture and Technology, Xi'an 710055, China;
    xinglu@live.xauat.edu.cn (L.X.); liuhuawei@xauat.edu.cn (H.L.); lvzhenyuan@live.xauat.edu.cn (Z.L.);
    zhuchao@xauat.edu.cn (C.Z.)
[2]   Xi'an Engineering Technology Research Center, Xi'an 710055, China; quanzonggang@xauat.edu.cn (Z.Q.);
    fuguangming@xauat.edu.cn (G.F.)
[3]   Xi'an Research & Design Institute of Wall and Roof Materials, Xi'an 710061, China
[4]   Shaanxi Key Laboratory of Safety and Durability of Concrete Structures, Xijing University,
    Xi'an 710123, China; wujian@xauat.edu.cn
\*   Correspondence: chaoliu@xauat.edu.cn; Tel.: +86-180-9256-1062

**Abstract:** In this paper, the bond deterioration mechanism of recycled aggregate concrete (RAC) with a full replacement ratio was studied through experimental and numerical simulations. To study the bond behavior and the bond slip between section steel and RAC, nine push-out specimens were designed using the control variable method. The effects of the concrete strength, the embedded length, the cover thickness, and the lateral stirrup ratio on the bond behavior and the bond slip were investigated in detail. The loading process and failure mode of the specimens were observed, and the test curves of the loading end and free end of the specimens were analyzed. Based on the experiment, the finite element method (FEM) was used to simulate the specimens, and the simulation results were analyzed by comparing the experiment data. The analysis of the results showed that the developed model is capable of representing the characteristic bond strength value between section steel and RAC with sufficient accuracy, and the main differences of bond slip between the simulation and the test results are the slippage at the limit state and the moment at which the free end starts to slip.

**Keywords:** full replacement ratio; section steel and RAC; bond behavior; SRRC (Steel Reinforced Recycled Concrete); bond strength; bond slip; numerical simulation

## 1. Introduction

With the acceleration of industrialization and urbanization, the construction industry has developed rapidly, and the accompanying construction waste has increased dramatically. Due to the large cost of traditional landfill methods and serious environmental pollution, reusing the construction waste resources is imperative [1,2]. As a product of the recycling construction waste, recycled aggregate concrete (RAC) has its own advantages in terms of economy and protection for the environment. RAC is prepared by mixing recycled aggregate [3–6] with a certain proportion and grading, and partially or completely replacing natural aggregates (mainly coarse aggregates). The recycled aggregates are obtained by crushing, cleaning, and classifying the waste concrete.

In recent years, various scholars have conducted investigations on the properties of RAC [7–13]. It was found that RAC with an optimized mix ratio has better mechanical properties compared with ordinary concrete [14,15]. RAC has low weight, which is beneficial for reducing the weight of structure

and improving the seismic performance of RAC members. Therefore, the combined application of RAC and section steel in construction engineering not only reduces the overall weight of a structure, but also solves the problem of poor bearing capacity and rigidity of the structure.

Previous investigations [16–23] indicated that the bond slip behavior between section steel and RAC is an important factor affecting the structural performance, and the numerical simulation method of bond slip is an important theoretical basis of engineering extension and structural calculation analysis. Chen et al. [24] carried out push-out tests on twenty-two specimens of RAC with different replacement rates, and analyzed the effects of the replacement rate, the cover thickness, the bonding site, the hooping ratio, and the particle diameter of the aggregate on the bond strength. However, a mathematical model of the constitutive relationship for bond slip was not proposed. Liu et al. [25] studied the bond slip performance of section steel and RAC under different replacement rates and carried out a push-out test on thirty-six specimens. The constitutive relationship of bond strength under the position function was obtained, and the relationship between replacement rate and average characteristic bond strength was also acquired. Hwang et al. [26] established a numerical model for the bond slip analysis of concrete-filled steel tubular columns, considering the bond slip effect at two nodes. Al-Rousan et al. [27] established a new bond slip model for fiber-reinforced concrete, characterized by the fact that the model was based on anchored carbon fiber rebar and fiber-reinforced concrete.

In general, the research studies on the bond slip and numerical simulation between section steel and RAC are incomplete, especially for section steel and RAC with a full replacement ratio. Further study is important for determining the theoretical and engineering relationships between section steel and RAC.

In this study, the impact factors of bond slip strength between section steel and RAC with a full replacement ratio and the constitutive relationship of bond slip were studied. Moreover, reliable numerical simulation was provided to solve engineering challenges and to be used by other researchers and engineers.

## 2. Experimental Programs

### 2.1. Materials

In the study, the RAC with full replacement ratio means that all of the natural coarse aggregates in the concrete have been replaced by recycled coarse aggregates. The specimens were obtained from PC32.5R Portland cement, recycled coarse aggregate, natural fine aggregate, natural fine sand, and urban tap water. The particle size of the recycled coarse aggregate was ranged 5–31.5 mm. The physical properties of the recycled coarse aggregate [28,29] were measured, and the results are shown in Table 1. The obtained results satisfied the requirements of "Recycled Coarse Aggregate for Concrete" GB/T 25177-2010.

**Table 1.** Physical properties of recycled coarse aggregate.

| Bulk Density (kg/m$^3$) | | Needle-Flaky Particle Content (%) | Crushing Indicator (%) | Apparent Density (kg/m$^3$) | Water Absorption Rate (%) |
|---|---|---|---|---|---|
| Close | Loose | | | | |
| 1430 | 1309 | 3.90% | 17 | 2458 | 3.83 |

Double 10 channel steel plates and double 6mm thick steel plates were bonded with epoxy resin in the experiment. The longitudinally stressed steel provided B16 reinforcement, the stirrup provided A6 reinforcement, and their mechanical performances are shown in Table 2.

**Table 2.** Mechanical performances of steel and stirrup.

| Steel | | $E_1$/MPa | $E_2$/MPa | $\varepsilon_y$ | $\varepsilon_s$ | $\varepsilon_u$ | $f_y$/MPa | $f_u$/MPa |
|---|---|---|---|---|---|---|---|---|
| 6 mm steel plate | | $2.06 \times 10^5$ | $0.975 \times 10^5$ | 1738 | 20,500 | 29,500 | 354 | 425 |
| 10 channel steel | Flange | $2.08 \times 10^5$ | $0.985 \times 10^5$ | 1690 | 21,500 | 29,000 | 357 | 420 |
| | web | $2.07 \times 10^5$ | $0.967 \times 10^5$ | 1755 | 21,500 | 28,500 | 348 | 435 |
| A6 reinforcement | | $2.03 \times 10^5$ | $0.857 \times 10^5$ | 1510 | 14,500 | 22,500 | 310 | 350 |
| B16 reinforcement | | $2.06 \times 10^5$ | $0.995 \times 10^5$ | 1895 | 16,500 | 30,500 | 390 | 530 |

Note: $E_1$ is the elastic modulus of the elastic phase; $E_2$ is the slope of the hardening section of the steel; $\varepsilon_y$ is the yield strain of the steel corresponding to $f_y$; $\varepsilon_s$ is the strain of the steel; $\varepsilon_u$ is the peak strain of the steel corresponding to $f_u$; $f_y$ is the yield strength of the steel; $f_u$ is the ultimate strength of the steel.

## 2.2. Design of Specimens

Nine push-out specimens were designed in the test to study the bond behavior and the bond slip between section steel and RAC. The effects of the concrete strength, the embedded length, the cover thickness, and the lateral stirrup ratio on the bond behavior and the bond slip between section steel and RAC were investigated in detail. The parameters of push-out specimens are shown in Table 3. All strain gauges were arranged at a certain interval on the flange steel plate and the web channel steel. This arrangement did not affect the bonding effect in the interface between section steel and RAC, and ensured the safety and the accuracy of the strain gauge. The section design of specimens and section steel are shown in Figures 1 and 2, respectively.

**Table 3.** Parameters of push-out specimens.

| No. | Concrete Strength | Embedded Length/mm | Cover Thickness/mm | Lateral Stirrup Ratio/% | Stirrup Configuration | $f_{cu}$/MPa |
|---|---|---|---|---|---|---|
| SRRC-1 | C30 | 740 | 55 | 0.2 | A6@140 | 29.9 |
| SRRC-2 | C30 | 540 | 55 | 0.2 | A6@140 | 32.43 |
| SRRC-3 | C30 | 740 | 40 | 0.2 | A6@160 | 32.43 |
| SRRC-4 | C30 | 740 | 70 | 0.2 | A6@120 | 32.43 |
| SRRC-5 | C40 | 740 | 55 | 0.2 | A6@140 | 45.12 |
| SRRC-6 | C20 | 740 | 55 | 0.2 | A6@140 | 21.60 |
| SRRC-7 | C30 | 740 | 55 | 0.25 | A6@110 | 29.9 |
| SRRC-8 | C30 | 740 | 55 | 0.3 | A6@95 | 29.9 |
| SRRC-9 | C30 | 940 | 55 | 0.2 | A6@140 | 29.9 |

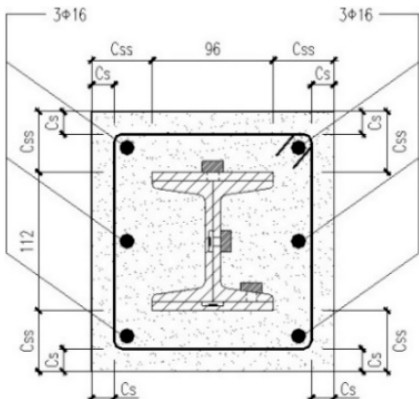

**Figure 1.** Section design of specimens.

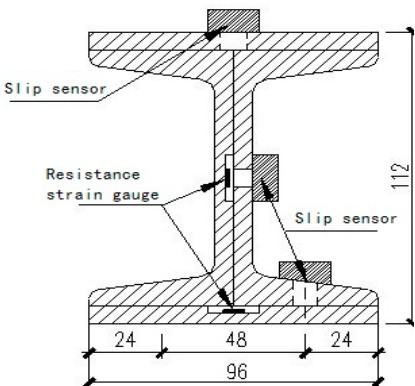

**Figure 2.** Section design of section steel.

Pre-absorption treatment was conducted on recycled coarse aggregate before preparing RAC [30,31], and the mix design of RAC is shown in Table 4. The reason is that the recycled coarse aggregate is largely porous, which reduces the actual water/cement ratio in the cement slurry and the concrete mix ratio at the same concrete strength.

**Table 4.** Mix design of recycled aggregate concrete (RAC).

| Concrete Strength | Dosage/Kg·m$^{-3}$ | | | | |
|---|---|---|---|---|---|
| | Cement | Recycled Coarse Aggregate | Sand | Water | Additional Water |
| C20 | 335.0 | 1250.0 | 650.0 | 195.0 | 48.0 |
| C30 | 370.0 | 1185.0 | 660.0 | 195.0 | 45.4 |
| C40 | 425.0 | 1225.0 | 640.0 | 145.0 | 46.9 |

Pre-absorption treatment was conducted on recycled coarse aggregate before preparing RAC [30,31]. The reason is that the recycled coarse aggregate with large porosity absorbs a lot of water, and decreases the actual water-cement ratio in the cement slurry. The mix design of RAC is shown in Table 4.

The specimens were made in the seismic engineering laboratory of Xi'an University of Architecture and Technology. The 150 mm × 150 mm × 150 mm cube test blocks were also produced from the same RAC in the test. After the pouring was completed, the push-out specimens were cured in indoor standard conditions (with felt-covered watering and curing). The compressive strength ($f_{cu}$) of specimens is shown in Table 3.

### 2.3. Test Method

The strain gauges were applied from dense to sparse distribution along the loading end to the free end, and were bonded to the steel plate by epoxy resin to measure the strain at the flange and the web. Four electronic slip sensors were uniformly arranged on one side of the flange and the web, which were developed by the research team [32], and the Slip-strain (S–$\varepsilon$) relationship of each electronic slip sensor was measured in advance.

The push-out test was carried out on the 2000t compression testing machine in the State Key Laboratory for civil engineering at Xi'an University of Architecture and Technology. Figure 3a,b show photos of the test setup and push-out specimen. The upper end of the specimen was fixed and the lower end was free. A mild steel plate with an "H" hole connected the bottom of the specimen with the loading platform. The topside of the section steel was attached to the compression testing machine with a complete steel plate. A foam pad was laid between the steel support and RAC to ensure flatness, and the loading rate was 0.3 mm/min. Slip occurred initially in the lower part and gradually in the upper part. Therefore, from the perspective of the section steel, the loading end of the specimen was defined at the lower end and the free end was in the upper portion. Double displacement meters were set at the loading end and the free end, respectively, which are shown in Figure 3c.

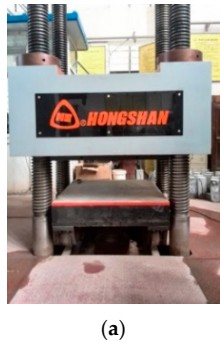
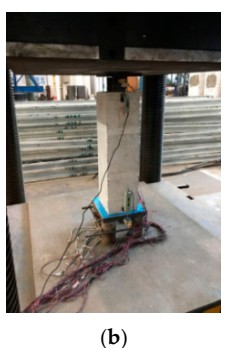
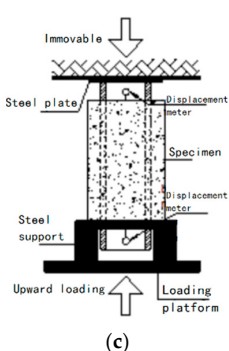

(**a**)　　　　　　　　　　　(**b**)　　　　　　　　　　　(**c**)

**Figure 3.** Photos and sketch of the test: (**a**) photo of test setup; (**b**) photo of push-out specimen; (**c**) sketch of push-out specimen.

## 3. Results and Analysis

### 3.1. Failure Procedure and Mode

The failure modes of the specimens can be divided into two types: splitting failure mode and bursting failure mode. The failure procedure was roughly as follows: at the initial stage of the specimen loading, there was no obvious change on the surface of each specimen. When the specimen was loaded to 40%–75% of the ultimate load, the initial cracks appeared on the surface of the specimen. With the increase of loading, the initial cracks were mostly concentrated near the loading end at the web, and a small part appeared in the middle. In this time, the initial cracks propagated rapidly, the initial cracks at the loading end extended toward the free end, and the initial cracks in the middle expanded toward both sides as the load increased. When loading to 80%–90% of the ultimate load, the sliding increment of the loading end was obvious and the load increased gently. The initial crack gradually developed into a through crack, and the maximum crack width reached 2–3 mm. As the load continued increasing to the ultimate load, the load sharply dropped to 50%–70% of it, therefore the specimen was considered to be broken by the through crack. If the load continued to increase, the changes were minimal and stabilized with the increasing drifts. It was considered that the load was a residual load. In the process, multiple cracks were generated, in which the original cracks developed secondary cracks, and the damage of RAC increased. The loading ended when the section steel was pushed out 4–6 mm.

The failure mode for SRRC-1, SRRC-4, and SRRC-8 was bursting failure, and the rest specimens showed splitting failure. Splitting failure is the typical failure mode, and here the initial cracks appeared at the loading end of the web sides. With increased load, the cracks extended to the free end and some fine cracks appeared on the specimen gradually. When the load reached the peak load, the initial cracks extended to the upper part of the specimen. Then the load fell rapidly and tended to be gentle gradually. There was a penetrating crack on the flange side and at web sites at this stage, as shown in Figure 4 (taking SRRC-5 as an example).

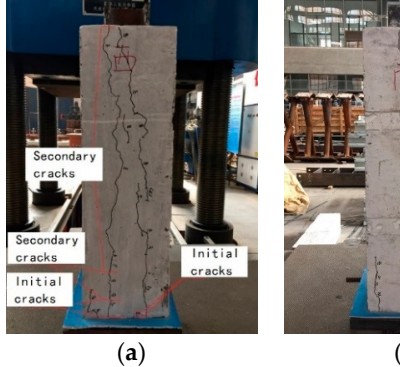
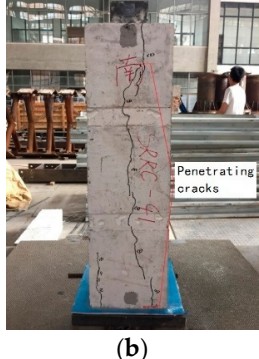

(**a**)　　　　　　　　　　　(**b**)

**Figure 4.** Splitting failure of SRRC-5: (**a**) web; (**b**) flange.

With bursting failure, the initial cracks occurred in the middle of the flange or the web. As the load increased, the initial cracks gradually expanded toward the loading and free ends, and some new fine cracks occurred. When the load reached about 80%–90% of the ultimate load, the initial cracks continued expanding and widening, and there were many obvious secondary cracks. As shown in Figure 5 (taking SRRC-1 as an example), through cracks were present on both the flange and the web sides after failure. This is one of the main features of bursting failure that makes it different from the splitting failure.

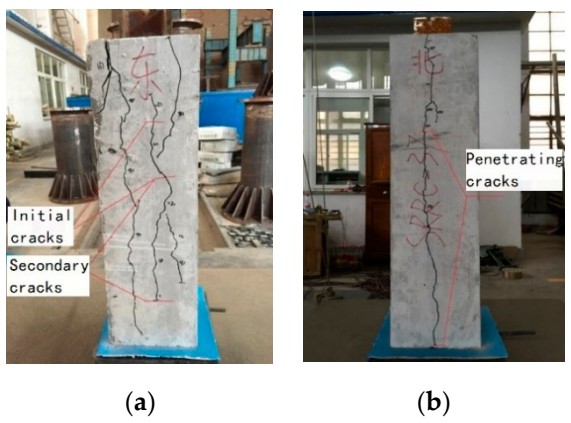

(**a**)　　　　　　　　(**b**)

**Figure 5.** Bursting failure of SRRC-1: (**a**) web; (**b**) flange.

It can be seen from SRRC-4 and SRRC-8 specimens that a high lateral stirrup ratio and high cover thickness make the specimen more prone to bursting failure. The reason for this phenomenon is that a high lateral stirrup ratio and high cover thickness are effective in preventing the deformation of concrete and further improving the cracking load of cracks.

*3.2. Characteristic of P–S Curves*

The loading end slip curve (P-S curve) can be simplified to the model shown in Figure 6. Here, The load is divided into two categories, each of them showing basically the same changes, which are divided into three parts: rising, sag, and gentle loads. Type (I) is characterized by a large initial load (65%–75% of the peak load), with a residual load that is slightly lower than the initial load. Type (II) is characterized by a small initial load (40%–65% of the peak load), with a residual load that is slightly higher than the initial load. The P–S curves of the specimens are shown in Figure 7.

The following definitions of the characteristic points in Figure 6 are given:

(1)　The initial load Ps: The load when obvious slippage occurred on specimens (point A)
(2)　The ultimate load Pu: The maximum value of the specimens (point B)
(3)　The residual load Pr: The load corresponding to the end of the descending stage (point C)

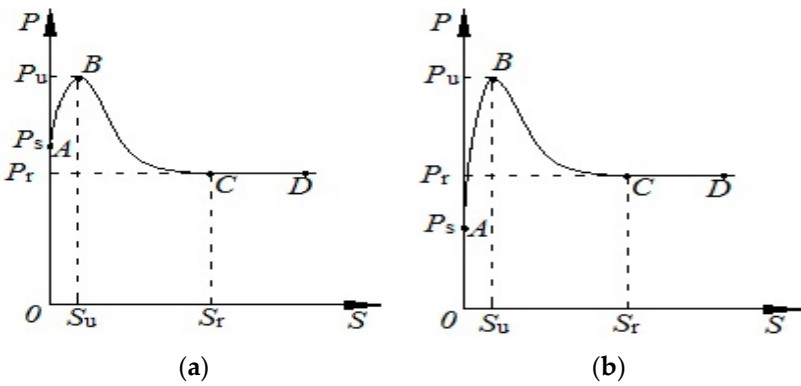

**Figure 6.** P–S curve models of the loading end: (**a**) Type (I); (**b**) Type (II).

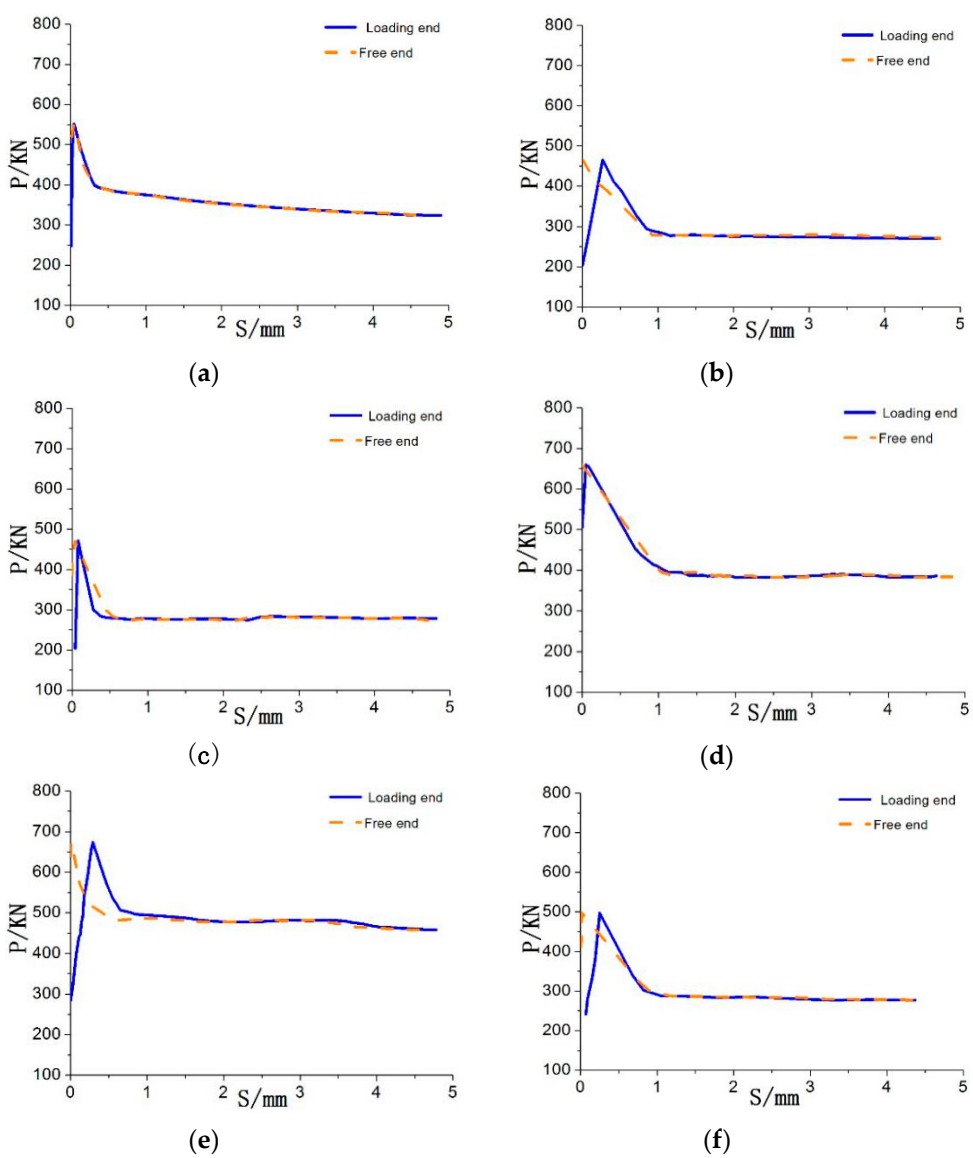

**Figure 7.** *Cont.*

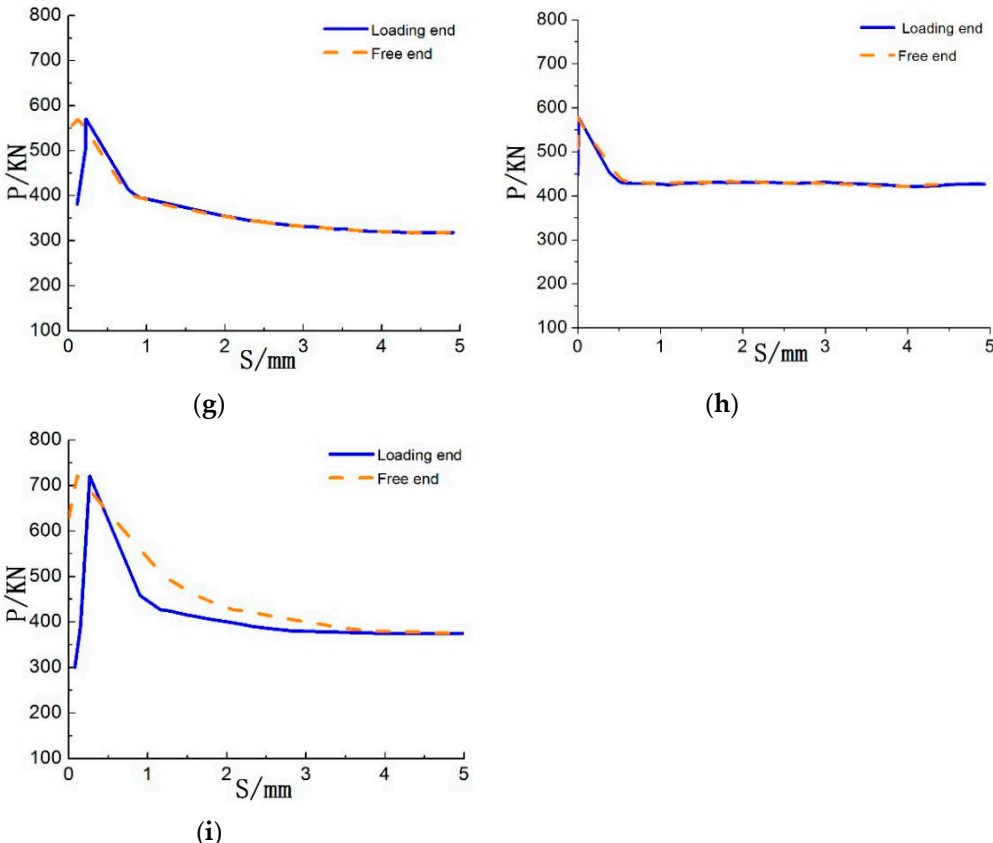

**Figure 7.** P–S curves of each specimen: (**a**) SRRC-1; (**b**) SRRC-2; (**c**) SRRC-3; (**d**) SRRC-4; (**e**) SRRC-5; (**f**) SRRC-6; (**g**) SRRC-7; (**h**) SRRC-8; (**i**) SRRC-9.

In this paper, the P–S curves of the loading end are divided into four stages: nonslip, slip-crack, descending, and residual.

(1)   OA in Figure 6 indicates the nonslip stage of the specimens. The key point is the initial bond load point, which determines the length of the section, and the main load is borne by the chemical adhesive force at this section. The composition of the bond stress in the section steel and RAC is similar to in the section steel and ordinary concrete. The bond stress is caused by chemical adhesion and frictional resistance in the article [33].

(2)   AB in Figure 6 indicates the slip-crack stage of the specimens, where the curve is basically a linear relationship and the slope is too large. When loading to 40%–65% of the ultimate load (the load is defined as the initial load $P_s$, the corresponding bond strength is the average initial bond strength $\bar{\tau}_s$), the loading end of the specimen begins to slip and developed rapidly.

(3)   The load drops sharply and the specimen has longitudinal through cracks when loading increases to the ultimate load $P_u$ (the corresponding bond strength is the average limited bond strength $\bar{\tau}_u$). The reason is that the chemical adhesion of the descending stage is suddenly broken and the friction is not sufficient to support the ultimate load.

(4)   The residual mainly depends on the residual load. The P–S curve is basically a horizontal line when the load falls to 50%–70% of the ultimate load (the load is defined as the residual load $P_r$, the corresponding bond strength is the average residual bond strength $\bar{\tau}_r$). It can be concluded that the determinants of each stage are the characteristic loads.

### 3.3. Influence Analysis of Various Factors

The bond strength between the section steel and RAC can be considered to be evenly distributed along the length of the section steel under the push-out test conditions. The average bond strength can be expressed by Equation (1).

$$\overline{\tau} = \frac{P}{L_e \cdot C} \tag{1}$$

where $\overline{\tau}$ is the average bond stress in MPa; $P$ is the load in N; $L_e$ is the embedded length of section steel in mm; and $C$ is the perimeter of section steel in mm.

### 3.3.1. Concrete Strength

The bond strength is basically a linear relationship with the tensile strength of RAC, and the bond strength increases with the increase of tensile strength, as shown in Figure 8a. Equation (2) [34] was adopted in this study, which reflects the relationship between the tensile strength and compressive strength of RAC.

$$f_t = 0.18 f_{cu}^{\frac{2}{3}} \tag{2}$$

where $f_{cu}$ is the compressive strength of RAC; $f_t$ is the tensile strength of RAC.

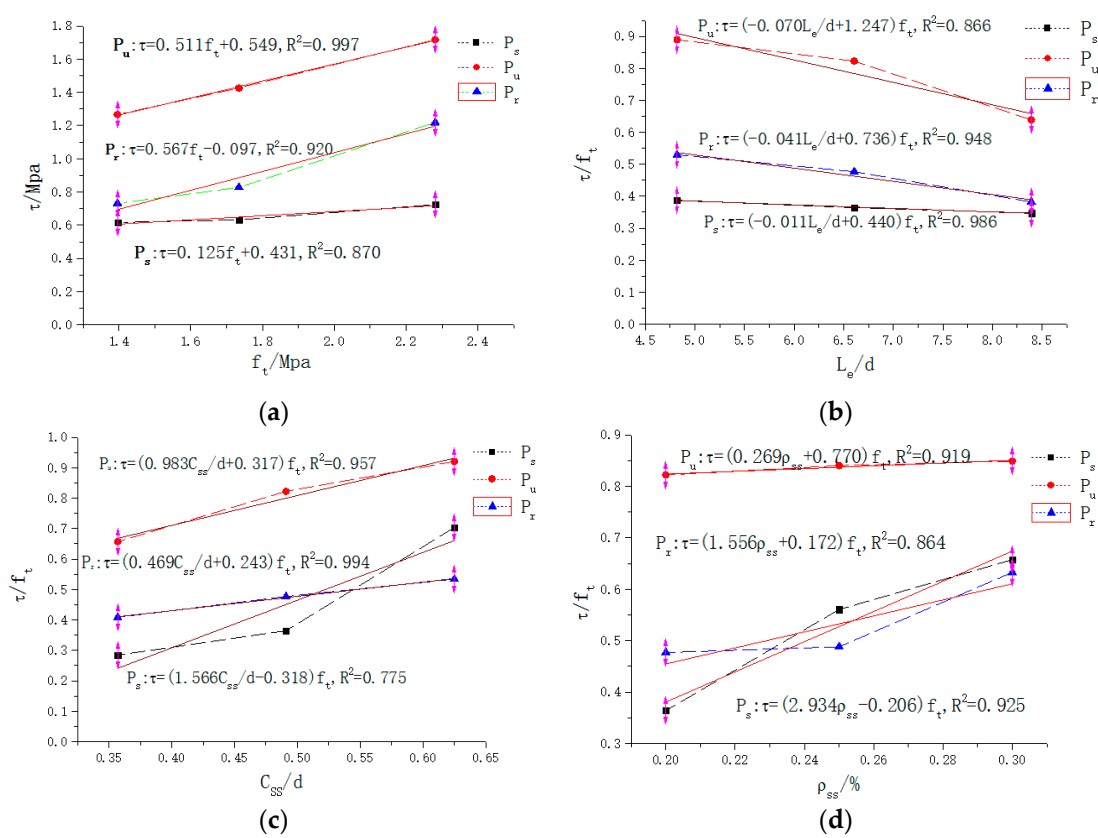

**Figure 8.** Relationship of bond strength and various factors: (**a**) concrete strength; (**b**) embedded length; (**c**) cover thickness; (**d**) lateral stirrup ratio.

The relationship between the tensile strength ($f_t$) of RAC and the average bond strength ($\overline{\tau}$) is obtained by statistical regression, which is fit as Equations (3)–(5).

$$\overline{\tau_s} = 0.125 f_t + 0.431 \tag{3}$$

$$\overline{\tau_u} = 0.511 f_t + 0.549 \tag{4}$$

$$\overline{\tau_r} = 0.567 f_t - 0.097 \tag{5}$$

### 3.3.2. Embedded Length

The relative bond strength is defined as the ratio of the average bond strength to the tensile strength ($\tau / f_t$), and the relative embedded length is defined as the ratio of the embedded length to the height of the section steel ($L_e / d$). The relationship between the embedded length and the average initial bond strength, the average ultimate bond strength, and the average residual bond strength are shown in the following equations:

$$\overline{\tau_s} = (-0.011 L_e / d + 0.440) f_t \tag{6}$$

$$\overline{\tau_u} = (-0.070 L_e / d + 1.247) f_t \tag{7}$$

$$\overline{\tau_r} = (-0.041 L_e / d + 0.736) f_t \tag{8}$$

As can be seen from Figure 8b, the bond stress decreases as the embedded length increases. The reduction effect of the average initial bond strength is not obvious, and the average ultimate bond strength decreases significantly.

### 3.3.3. Cover Thickness

The relative cover thickness is calculated from the ratio of the cover thickness to the height of the section steel ($C_{ss}/d$). The cover thickness refers to the distance between the section steel and the outer surface of RAC. The relationship is shown in the following equations:

$$\overline{\tau_s} = (1.566 C_{ss} / d - 0.318) f_t \tag{9}$$

$$\overline{\tau_u} = (0.983 C_{ss} / d + 0.317) f_t \tag{10}$$

$$\overline{\tau_r} = (0.469 C_{ss} / d + 0.243) f_t \tag{11}$$

It can be seen from Figure 8c that the average characteristic bond strength obviously increases with the increase of the cover thickness.

### 3.3.4. Lateral Stirrup Ratio

The effect of the lateral stirrup ratio is similar to that of the cover thickness, which can effectively prevent the lateral deformation of the RAC and delay the cracking time. The equations are as follows.

$$\overline{\tau_s} = (2.934 \rho_{ss} - 0.206) f_t \tag{12}$$

$$\overline{\tau_u} = (0.269 \rho_{ss} - 0.770) f_t \tag{13}$$

$$\overline{\tau_r} = (1.556 \rho_{ss} - 0.172) f_t \tag{14}$$

It can be seen from Figure 8d that the average characteristic bond strength increases with the increase of the lateral stirrup ratio, and the average initial bond strength increases obviously. The effect of increasing the ultimate bond strength is poor, indicating that the increase of the lateral stirrup ratio can effectively delay the appearance of the initial crack and increase the cracking load, but the effect on improving the average ultimate bond strength is not significant.

*3.4. Formulas*

The characteristic bond load and the average characteristic bond strength of specimens are shown in Table 5. The formulas for the average bond stress of the four factors were established by statistical regression analysis. They can be expressed as follows.

$$\overline{\tau}_s = (\frac{-0.686C_{ss}}{d} + \frac{0.020L_e}{d} + 2.506\rho_{sv} + 0.067)f_t \tag{15}$$

$$\overline{\tau}_u = (\frac{-0.335C_{ss}}{d} + \frac{0.015L_e}{d} + 0.718\rho_{sv} + 0.683)f_t \tag{16}$$

$$\overline{\tau}_r = (\frac{-0.493C_{ss}}{d} + \frac{0.006L_e}{d} - 0.842\rho_{sv} + 0.590)f_t \tag{17}$$

where $\overline{\tau}_s$ is average initial bond strength; $\overline{\tau}_u$ is average ultimate bond strength; $\overline{\tau}_r$ is average residual bond strength.

In order to verify the reliability of the formulas, the comparison was performed between the calculation of the formulas and the experiment data from this test, as well as using data from Yin et al. [35], Chen et al. [24], and Chen et al. [36]. The results are shown in Table 6.

Table 6 indicates that the average ultimate bond strength and the average residual bond strength fit well, but the fitting result of the average initial bond strength has a certain error. One of the reasons for this error is the different values of initial load between the man-made and instrument methods. In addition, Equation (2) by Xiao et al. [34] for the tensile strength of RAC was used in this study, but the rest of the articles adopted ordinary concrete formulas. The tensile strength of RAC under the same compressive strength is higher than in this paper.

**Table 5.** Results of characteristic load and bond stress tests.

| No. | Main Anchoring Condition | | | | | $P_s$/KN | $P_u$/KN | $P_r$/KN | $\bar{\tau}_s$/MPa | $\bar{\tau}_u$/MPa | $\bar{\tau}_r$/MPa |
|---|---|---|---|---|---|---|---|---|---|---|---|
| | Replacement (%) | Concrete Strength (MPa) | Cover Thickness (mm) | Embedded Length (mm) | Lateral Stirrup Ratio (%) | | | | | | |
| SRRC-1 | 100 | C30 | 55 | 740 | 0.2 | 248 | 559 | 324 | 0.633 | 1.426 | 0.827 |
| SRRC-2 | 100 | C30 | 55 | 540 | 0.2 | 203 | 466 | 278 | 0.710 | 1.629 | 0.972 |
| SRRC-3 | 100 | C30 | 40 | 740 | 0.2 | 204 | 471 | 275 | 0.520 | 1.202 | 0.702 |
| SRRC-4 | 100 | C30 | 70 | 740 | 0.2 | 505 | 660 | 383 | 1.288 | 1.684 | 0.977 |
| SRRC-5 | 100 | C40 | 55 | 740 | 0.2 | 284 | 673 | 478 | 0.724 | 1.717 | 1.219 |
| SRRC-6 | 100 | C20 | 55 | 740 | 0.2 | 242 | 497 | 287 | 0.617 | 1.268 | 0.732 |
| SRRC-7 | 100 | C30 | 55 | 740 | 0.25 | 381 | 571 | 332 | 0.972 | 1.457 | 0.847 |
| SRRC-8 | 100 | C30 | 55 | 740 | 0.3 | 447 | 577 | 430 | 1.140 | 1.472 | 1.097 |
| SRRC-9 | 100 | C30 | 55 | 940 | 0.2 | 300 | 721 | 376 | 0.602 | 1.448 | 0.755 |

**Table 6.** Comparison of characteristic bond strength.

| Source | No. | Initial Bond Strength | | Calculated/Tested | Limit Bond Strength | | Calculated/Tested | Residual Bond Strength | | Calculated/Tested |
|---|---|---|---|---|---|---|---|---|---|---|
| | | Calculated | Tested | | Calculated | Tested | | Calculated | Tested | |
| Article | SRRC-1 | 0.630 | 0.633 | 0.996 | 1.320 | 1.426 | 0.926 | 0.827 | 0.827 | 1.000 |
| | SRRC-2 | 0.600 | 0.710 | 0.845 | 1.344 | 1.629 | 0.825 | 0.892 | 0.972 | 0.918 |
| | SRRC-3 | 0.833 | 0.520 | 1.602 | 1.475 | 1.202 | 1.228 | 0.993 | 0.702 | 1.416 |
| | SRRC-4 | 0.497 | 1.288 | 0.386 | 1.311 | 1.684 | 0.779 | 0.751 | 0.977 | 0.769 |
| | SRRC-5 | 0.829 | 0.724 | 1.144 | 1.736 | 1.717 | 1.011 | 1.087 | 1.219 | 0.892 |
| | SRRC-6 | 0.507 | 0.617 | 0.822 | 1.063 | 1.268 | 0.838 | 0.665 | 0.732 | 0.909 |
| | SRRC-7 | 0.848 | 0.972 | 0.872 | 1.382 | 1.457 | 0.949 | 0.900 | 0.847 | 1.062 |
| | SRRC-8 | 1.065 | 1.140 | 0.934 | 1.444 | 1.472 | 0.981 | 0.973 | 1.097 | 0.887 |
| | SRRC-9 | 0.731 | 0.602 | 1.213 | 1.442 | 1.448 | 0.996 | 0.853 | 0.755 | 1.129 |
| Chen et al. | SRRAC-11 | 1.438 | 1.670 | 0.861 | 1.719 | 1.963 | 0.876 | 1.258 | 1.404 | 0.896 |
| Chen et al. | SRAC-5 | 1.318 | 0.930 | 1.417 | 1.821 | 1.510 | 1.206 | 1.238 | 0.990 | 1.251 |
| Yin et al. | SRRC-34 | 0.860 | 0.501 | 1.717 | 1.213 | 0.905 | 1.340 | 0.801 | 0.412 | 1.944 |
| | SRRC-35 | 0.807 | 0.349 | 2.311 | 1.125 | 1.139 | 0.988 | 0.798 | 0.629 | 1.269 |
| | SRRC-36 | 0.379 | 0.492 | 0.770 | 1.000 | 1.038 | 0.964 | 0.573 | 0.637 | 0.899 |

## 4. Numerical Simulation

The simulation of interfacial bond stress is a difficult point in the simulated process of bond slip between section steel and RAC. Nonlinear spring units were utilized to solve the problem in this study, which included two aspects: one was the preparation of the nonlinear syntax for the inp file, and the other was the determination of the constitutive relationship of the spring element.

### 4.1. Finite Element Model

### 4.1.1. Element and Material

The solid element C3D8R for section steel and RAC was used in this study, which is an 8-node hexahedron reduction integral element. This element has more accurate results and saves calculation time, and is also suitable for meshing refinement. The linear three-dimensional truss element T3D2 was adopted for steel and stirrups, which has two nodes, each with three degrees of freedom. The steel and stirrups were assigned as truss elements when meshing.

The experimental materials adopted in this study were described as shown in Table 7. The properties of the steel used in the tests were determined in accordance with the "Code for Design of Concrete Structures" [37]. The elastic properties of the second-class coarse aggregate for RAC are shown in Table 8. The plastic damage model of Abaqus was selected for the plastic part of the RAC.

**Table 7.** Properties of steel.

| Material | Elastic Modulus/MPa | Density/Kg/m$^3$ | Poisson's Ratio | Plasticity | |
|---|---|---|---|---|---|
| | | | | Yield Stress/MPa | Plastic Strain |
| 6 mm Steel plate No. 10 channel steel | 206,000 | 7850 | 0.3 | 354 | 0 |
| A6 Reinforcement (HPB300) | 203,000 | 7850 | 0.3 | 310 | 0 |
| B16 Reinforcement (HRB335) | 206,000 | 7850 | 0.3 | 390 | 0 |

**Table 8.** Properties of concrete.

| Concrete Strength | Elastic Modulus/MPa | Density/Kg/m$^3$ | Poisson's Ratio |
|---|---|---|---|
| C20 | 18,480 | 1700 | 0.2 |
| C30 | 23,420 | 1700 | 0.2 |
| C40 | 28,510 | 1700 | 0.2 |

### 4.1.2. Analysis Step and Constraint

The initial incremental step was 0.02, the minimum incremental step was the default 0.00001, and the maximum incremental step was 1000. These were set to meet the calculation requirements in this simulation. In the interaction module, the embedding relationship was defined between the RAC and reinforcement cage, which was made of stirrups and steel. The reference point above the surface of the section steel acting on the displacement load was provided. The loading speed was 0.3 mm/min, and the loading frequency was set as the amplitude. The binding constraints between the bottom slab and the RAC were defined, and the boundary conditions at the bottom of the slab were set as fixed. The specimen adding constraint is shown in Figure 9.

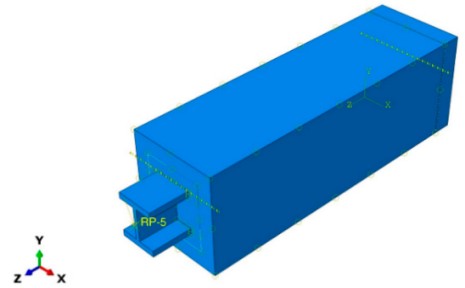

**Figure 9.** Specimen containing section steel and RAC.

### 4.1.3. Meshing

The mesh generation of the specimen was performed after the assembly of the components and the constraint settings were completed, which affected the establishment of subsequent nonlinear springs. The intersection interface for the spring element arrangement is most important, and it is located between the section steel and the RAC. The mesh for the interface between the section steel and the RAC was divided in order to successfully arrange the subsequent spring elements. The section steel was cut according to the geometrical axis, as shown in Figure 10, and the section steel and RAC parts were set as independent.

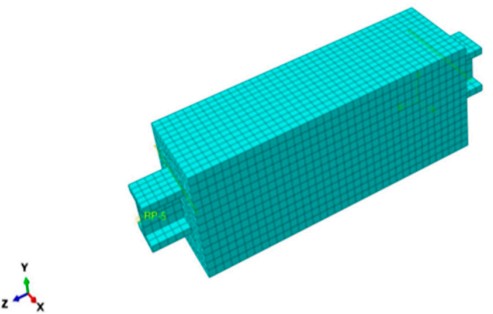

**Figure 10.** Mesh division for specimen containing section steel and RAC.

The node set of the interface for the section steel and the RAC was established after the mesh was divided, as shown in Figure 11. All nodes in the node set were exported in post-processing and were numbered using the "VLOOKUP" function in excel software.

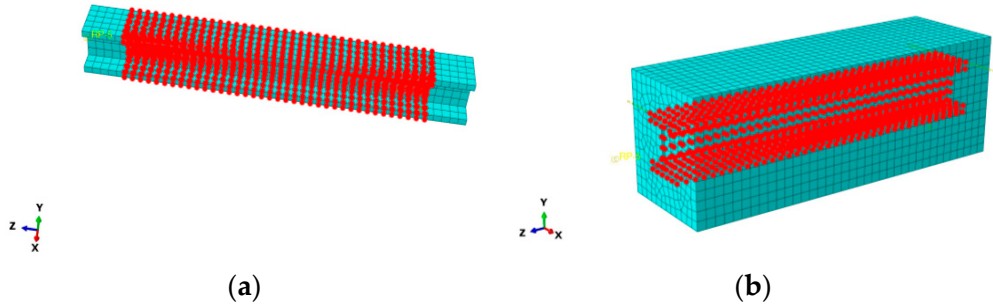

| (**a**) | (**b**) |
| --- | --- |

**Figure 11.** Node set: (**a**) section steel; (**b**) RAC.

Each specimen in this simulation has 1140 nodes. The VLOOKUP function of a corresponding node is

=VLOOKUP(C4&D4&E4,CHOOSE({1,2,3,4},$G$4:$G$1143&$H$4:$H$1143&$I$4:$I$1143,$F$4:$F$1143),2,0).

SRRC-1 is used as an example in Table 9.

**Table 9.** Relationship of node for SRRC-1.

| No. | Section 1 | Node 1 | Section 2 | Node 2 | No. | Section 1 | Node 1 | Section 2 | Node 2 |
|---|---|---|---|---|---|---|---|---|---|
| 1 | Tong-1 | 1 | Xinggang-1 | 15 | 42 | Tong-1 | 173 | Xinggang-1 | 283 |
| 2 | Tong-1 | 2 | Xinggang-1 | 14 | 43 | Tong-1 | 178 | Xinggang-1 | 1116 |
| 3 | Tong-1 | 3 | Xinggang-1 | 13 | 44 | Tong-1 | 179 | Xinggang-1 | 792 |
| 4 | Tong-1 | 4 | Xinggang-1 | 24 | 45 | Tong-1 | 180 | Xinggang-1 | 791 |
| 5 | Tong-1 | 5 | Xinggang-1 | 23 | 46 | Tong-1 | 181 | Xinggang-1 | 790 |
| 6 | Tong-1 | 6 | Xinggang-1 | 22 | 47 | Tong-1 | 182 | Xinggang-1 | 789 |
| 7 | Tong-1 | 7 | Xinggang-1 | 21 | 48 | Tong-1 | 183 | Xinggang-1 | 1296 |
| 8 | Tong-1 | 8 | Xinggang-1 | 20 | 49 | Tong-1 | 184 | Xinggang-1 | 1224 |
| 9 | Tong-1 | 9 | Xinggang-1 | 19 | 50 | Tong-1 | 185 | Xinggang-1 | 1223 |
| 10 | Tong-1 | 10 | Xinggang-1 | 18 | 51 | Tong-1 | 186 | Xinggang-1 | 1222 |
| 11 | Tong-1 | 11 | Xinggang-1 | 17 | 52 | Tong-1 | 187 | Xinggang-1 | 1260 |
| 12 | Tong-1 | 12 | Xinggang-1 | 16 | 53 | Tong-1 | 188 | Xinggang-1 | 936 |
| 13 | Tong-1 | 17 | Xinggang-1 | 499 | 54 | Tong-1 | 189 | Xinggang-1 | 935 |
| 14 | Tong-1 | 18 | Xinggang-1 | 516 | 55 | Tong-1 | 190 | Xinggang-1 | 934 |
| 15 | Tong-1 | 19 | Xinggang-1 | 515 | 56 | Tong-1 | 191 | Xinggang-1 | 933 |
| 16 | Tong-1 | 20 | Xinggang-1 | 514 | 57 | Tong-1 | 192 | Xinggang-1 | 972 |
| 17 | Tong-1 | 21 | Xinggang-1 | 513 | 58 | Tong-1 | 193 | Xinggang-1 | 1080 |
| 18 | Tong-1 | 22 | Xinggang-1 | 512 | 59 | Tong-1 | 194 | Xinggang-1 | 1079 |
| 19 | Tong-1 | 23 | Xinggang-1 | 511 | 60 | Tong-1 | 195 | Xinggang-1 | 1078 |
| 20 | Tong-1 | 24 | Xinggang-1 | 510 | 61 | Tong-1 | 323 | Xinggang-1 | 320 |
| 21 | Tong-1 | 25 | Xinggang-1 | 509 | 62 | Tong-1 | 324 | Xinggang-1 | 356 |
| 22 | Tong-1 | 26 | Xinggang-1 | 508 | 63 | Tong-1 | 325 | Xinggang-1 | 137 |
| 23 | Tong-1 | 27 | Xinggang-1 | 507 | 64 | Tong-1 | 326 | Xinggang-1 | 68 |
| 24 | Tong-1 | 28 | Xinggang-1 | 506 | 65 | Tong-1 | 327 | Xinggang-1 | 140 |
| 25 | Tong-1 | 29 | Xinggang-1 | 505 | 66 | Tong-1 | 328 | Xinggang-1 | 461 |
| 26 | Tong-1 | 30 | Xinggang-1 | 504 | 67 | Tong-1 | 329 | Xinggang-1 | 392 |
| 27 | Tong-1 | 31 | Xinggang-1 | 503 | 68 | Tong-1 | 330 | Xinggang-1 | 497 |
| 28 | Tong-1 | 32 | Xinggang-1 | 502 | 69 | Tong-1 | 331 | Xinggang-1 | 245 |
| 29 | Tong-1 | 33 | Xinggang-1 | 501 | 70 | Tong-1 | 332 | Xinggang-1 | 176 |
| 30 | Tong-1 | 34 | Xinggang-1 | 500 | 71 | Tong-1 | 333 | Xinggang-1 | 248 |
| 31 | Tong-1 | 162 | Xinggang-1 | 319 | 72 | Tong-1 | 334 | Xinggang-1 | 284 |
| 32 | Tong-1 | 163 | Xinggang-1 | 355 | 73 | Tong-1 | 339 | Xinggang-1 | 1115 |
| 33 | Tong-1 | 164 | Xinggang-1 | 138 | 74 | Tong-1 | 340 | Xinggang-1 | 788 |
| 34 | Tong-1 | 165 | Xinggang-1 | 67 | 75 | Tong-1 | 341 | Xinggang-1 | 787 |
| 35 | Tong-1 | 166 | Xinggang-1 | 139 | 76 | Tong-1 | 342 | Xinggang-1 | 786 |
| 36 | Tong-1 | 167 | Xinggang-1 | 462 | 77 | Tong-1 | 343 | Xinggang-1 | 785 |
| 37 | Tong-1 | 168 | Xinggang-1 | 391 | 78 | Tong-1 | 344 | Xinggang-1 | 1295 |
| 38 | Tong-1 | 169 | Xinggang-1 | 498 | 79 | Tong-1 | 345 | Xinggang-1 | 1221 |
| 39 | Tong-1 | 170 | Xinggang-1 | 246 | 80 | Tong-1 | 346 | Xinggang-1 | 1220 |
| 40 | Tong-1 | 171 | Xinggang-1 | 175 | 81 | Tong-1 | 347 | Xinggang-1 | 1219 |
| 41 | Tong-1 | 172 | Xinggang-1 | 247 | 82 | Tong-1 | 348 | Xinggang-1 | 1259 |

Note: Table cells are expressed in absolute form; C4, D4, and E4 are the columns where the x, y, and z coordinates of Section 1 are located. G4: G1143 represents the query columns, which correspond to the C4 column of Section 1; the rest can be deduced by analogy. F4: F1143 is the column for the output formula.

### 4.1.4. Plastic Damage Model

The premise for studying the plastic damage model for RAC is reflected in the constitutive relationship. The constitutive relationship between the compression and tension in RAC is basically the same as that of ordinary concrete. The specific differences are reflected in several coefficients related to the replacement rate. The equation proposed by Xiao et al. [38] was used for calculation in this study. The stress and strain at different strength levels are shown in Table 10.

**Table 10.** Characteristic parameters of RAC.

| Concrete Strength | Compressive Strength $f_c$/MPa | $\varepsilon_c$ | Tensile Strength $f_t$/Mpa | $\varepsilon_t$ | Elastic Modulus/MPa |
|---|---|---|---|---|---|
| C20 | 16.408 | 0.00124 | 1.396 | 0.00008 | 18,480 |
| C30 | 24.647 | 0.00147 | 1.830 | 0.00010 | 23,420 |
| C40 | 34.291 | 0.00168 | 2.281 | 0.00011 | 28,510 |

The plastic damage model of concrete is mainly used to provide a universal analysis model for analyzing the structure of concrete under cyclic and dynamic loads. This model is based on plastic and isotropic failure assumptions, and it can be used in unidirectional loading, cyclic loading, and other functions [39].

The evolution of the yield or failure surface is controlled by $\widetilde{\varepsilon}_c^{in}$ and $\widetilde{\varepsilon}_t^{in}$, where $\widetilde{\varepsilon}_c^{in}$ represents a compressive inelastic strain and $\widetilde{\varepsilon}_t^{in}$ represents a tensile inelastic strain.

$$\widetilde{\varepsilon}_c^{in} = \varepsilon_c - \varepsilon_{0c}^{el} \tag{18}$$

$$\varepsilon_{0c}^{el} = \sigma_c / E_0 \tag{19}$$

Let the plastic strain in the inelastic strain be $\widetilde{\varepsilon}_c^{pl}$, then the proportion of the inelastic strain is $\beta_c$, according to the Abaqus user manual:

$$\widetilde{\varepsilon}_c^{pl} = \widetilde{\varepsilon}_c^{in} - \frac{d_c}{1 - d_c} \times \frac{\sigma_c}{E_0} \tag{20}$$

$$d_c = \frac{(1 - \beta_C)\widetilde{\varepsilon}_c^{in} E_0}{\sigma_c + (1 - \beta_c)\widetilde{\varepsilon}_c^{in} E_0} \tag{21}$$

$$d_t = \frac{(1 - \beta_t)\widetilde{\varepsilon}_t^{in} E_0}{\sigma_t + (1 - \beta_t)\widetilde{\varepsilon}_t^{in} E_0} \tag{22}$$

where $d_c$ is the concrete compression damage factor; $d_t$ is the concrete tensile damage factor; $\sigma_c$ is the peak compressive stress of RAC; $\sigma_t$ is the peak tensile stress of RAC; $\beta_c$ is 0.6; $\beta_t$ is 0.8; $\widetilde{\varepsilon}_c^{in}$ is the inelastic compressive strain of RAC; $\widetilde{\varepsilon}_t^{in}$ is the inelastic tensile strain of RAC.

### 4.1.5. F–D Relation of Nonlinear Spring Unit

The constitutive relationship of the spring unit was determined before the establishment of the nonlinear spring unit, and was determined according to the constitutive relationship of the average bond strength at the loading end of the specimen. The interface between the section steel and RAC had three directions, namely longitudinal tangential, normal, and transverse tangential directions. The experiments proved that in the case of structural failure: first, the normal and transverse tangential deformation were much smaller than the longitudinal tangential deformation; second, the longitudinal tangential interaction was characterized by the bond slip phenomenon in the section steel and RAC. The Force-Displacement curve (F–D curve) consistent with the longitudinal tangential direction was employed by the spring element constitutive relationship, because the transverse tangential assumption was consistent with the longitudinal tangential interaction. The law-up interaction was set to a spring element with infinite stiffness because it was subjected to pressure and had high stiffness. The spring element F–D relationship is calculated by:

$$F = \tau \times A \tag{23}$$

where A is the area occupied the spring connection surface, with the calculation diagram shown in Figure 12.

The F–D relationship of the springs under each node can be calculated through the bond slip constitutive relation, which was obtained from the test. However, the calculated F–D relationship did not pass through the coordinate origin and did not satisfy the input requirements of Abaqus, so it was processed. It was completely symmetrically processed in its negative direction in order to complete the F–D curve. The F–D relationship is shown in Figure 13 (taking the intermediate node of the spring at SRRC-1 as an example).

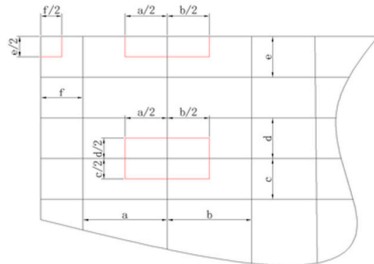

**Figure 12.** Schematic diagram of the force area calculation for the spring element.

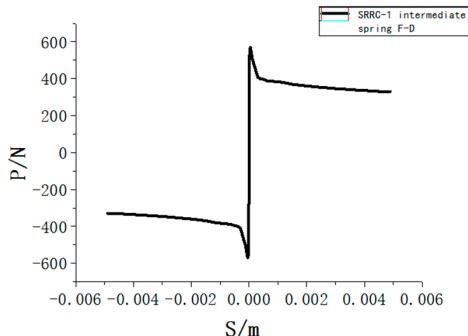

**Figure 13.** F–D curve for intermediate node spring element of SRRC-1.

### 4.1.6. Rewriting of Inp File

In this paper, the specific method used in addition of the nonlinear spring unit was as follows: the linear spring was first added to the simulated specimen, then the keywords in the inp file were found, and finally the linear spring was rewritten. The precautions were as follows: first, the rewritten inp file could not be imported into Computer Aided Engineering(CAE) and was applied by Abaqus command operation; second, the added maximum spring force was greater than the maximum force balanced with it; third, the nonlinear stiffness was symmetrically defined in the case of convenience and without affecting the result.

Taking SRRC-1 as an example, the nonlinear spring unit of the rewritten inp file is as follows:

*Spring, elset=Springs/Dashpots-1-spring, nonlinear
1, 1
Nonlinear constitutive relation
*Element, type=Spring2,
elset=Springs/Dashpots-1-spring
Serial number, part one, node one, part two, node two

Nonlinear constitutive relations and corresponding nodes were replaced in order to reduce the length of the article, as shown above. It should be noted that "nonlinear" was required as a keyword after the spring set, indicating that all springs in the set were nonlinear. Two points should be noted when adding nonlinear constitutive relations: one is that the F–D curve needs to pass the coordinate origin, and the other is that the force is connected by "," in the middle of the displacement. For example, in 570, 0.00005, 570 means the force is 570N, and 0.00005 means 0.00005 m. When adding multiple sets of spring sets, the serial number of the corresponding node should be accumulated at once, otherwise the nonlinear stiffness will be overwritten by subsequent coverage, resulting in the failure of multiple sets for spring addition.

In this paper, each set of springs had a total of 1140 nodes and each specimen contained three sets of springs, which were longitudinal tangential, transverse tangential, and normal. Nine sets of specimens were simulated and the results were as outlined in the following subsections.

### 4.2. Analysis

The model of the nine specimens was simulated. Taking SRRC-1 as an example, the Mises stress nephogram of the section steel and RAC are shown in Figure 14. It can be seen that the normal stress of the section steel gradually decreased from the loading end to the free end, because the stress point was directly coupled with the section steel. The loading end of the RAC is connected to the section steel only through a nonlinear spring, and the free end is directly contact with the steel plate. Therefore, for the RAC, the free end is directly stressed and the normal stress gradually decreases from the free end to the loading end.

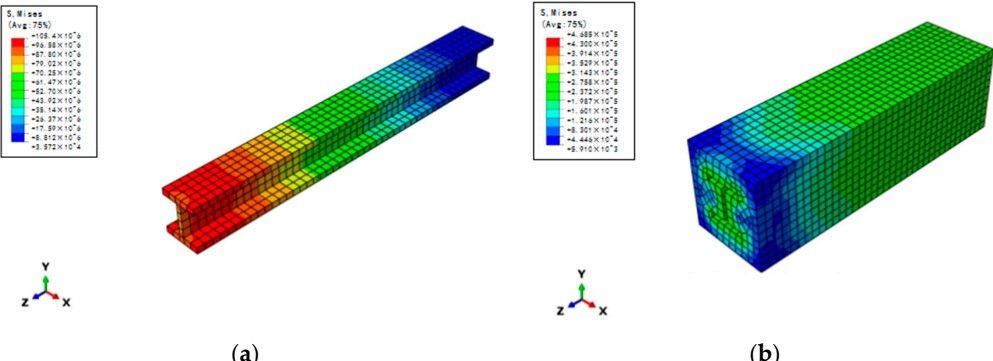

| (a) | (b) |
|---|---|

**Figure 14.** Mises stress nephogram of SRRC-1: (**a**) section steel; (**b**) RAC.

This study mainly investigated longitudinal shear stress. The stress nephograms of the specimen in the direction of S23 (S23 stands for shear stress in Abaqus) are shown in Figures 15 and 16. It can be seen that the longitudinal shear stress is relatively uniform distribution along the embedded length. The bond strength is weakest at the outside of the lower flange, and it is greatest at the inside of the lower flange and at the upper flange, which are the main bearing parts of the flange. The bond strength at the inner side of the web is equivalent to the outer side of the lower flange, which is much smaller than the bond strength at the outer side of the web.

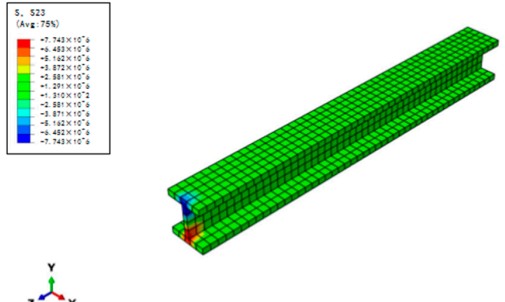

**Figure 15.** The stress nephograms of section steel in the direction of S23.

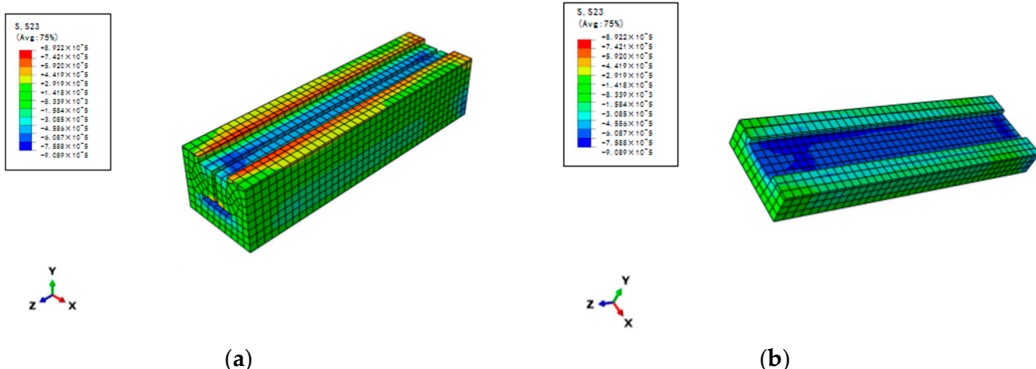

| (a) | (b) |
|---|---|

**Figure 16.** *Cont.*

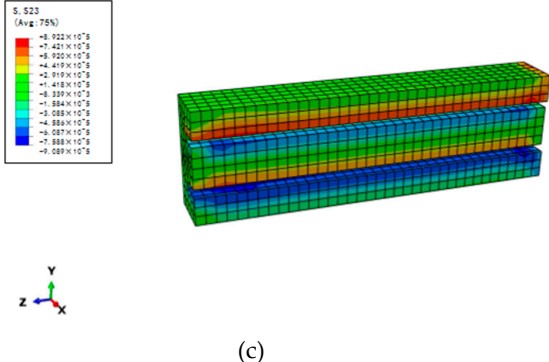

(c)

**Figure 16.** The cutting stress nephograms of RAC at different directions along the section steel: (**a**) the lower flange; (**b**) the upper flange; (**c**) the web.

According to the stress nephograms, the main failure surfaces in the bond slip of the specimens are the upper flange of the section steel, the inner side of the lower flange, and the outer side of the web. It is added that the bond strength of the outer side of the lower flange and the inner side of the web is much smaller than in the above three sides. Therefore, a certain process must be carried out on the outer side of the section steel flange and the inner side of the web, such as increasing the contact reaction area with the RAC and increasing the mechanical bite force with the RAC. The purpose of this is to enhance the characteristic bond strength of the specimen.

### 4.2.1. Comparison

As shown in Figure 17, the simulation figure is compared with the experimental figure. It can be seen that the simulation curves are basically consistent with the test curves. The initial load is basically equal with the ultimate load. The residual load has a slight error, which is within ±0.2 MPa and is reasonable. This is consistent with the test showing that SRRC-4, SRRC-7, and SRRC-8 belong to Type (I), with the rest of the specimens belonging to Type (II).

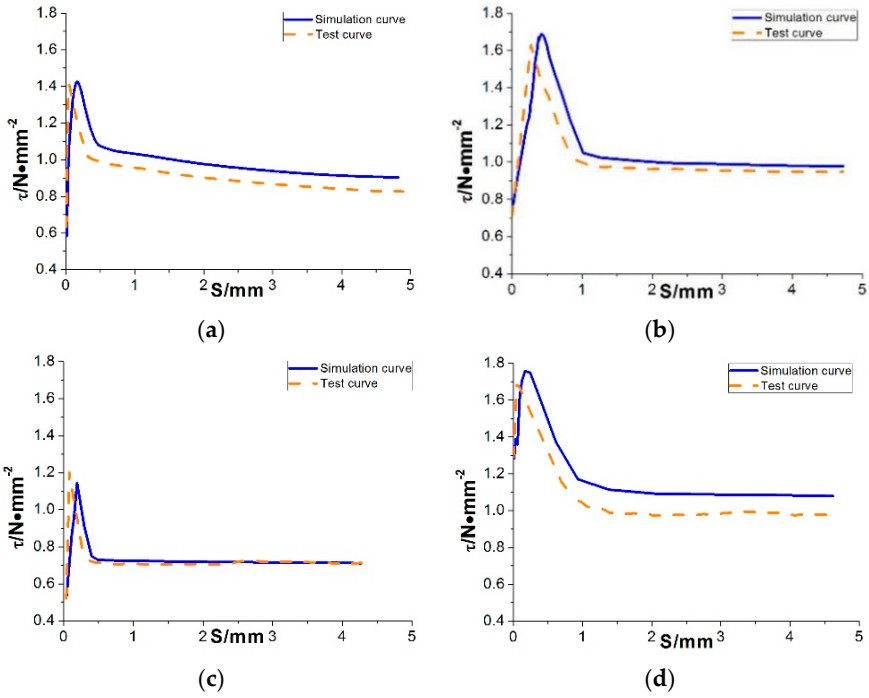

**Figure 17.** *Cont.*

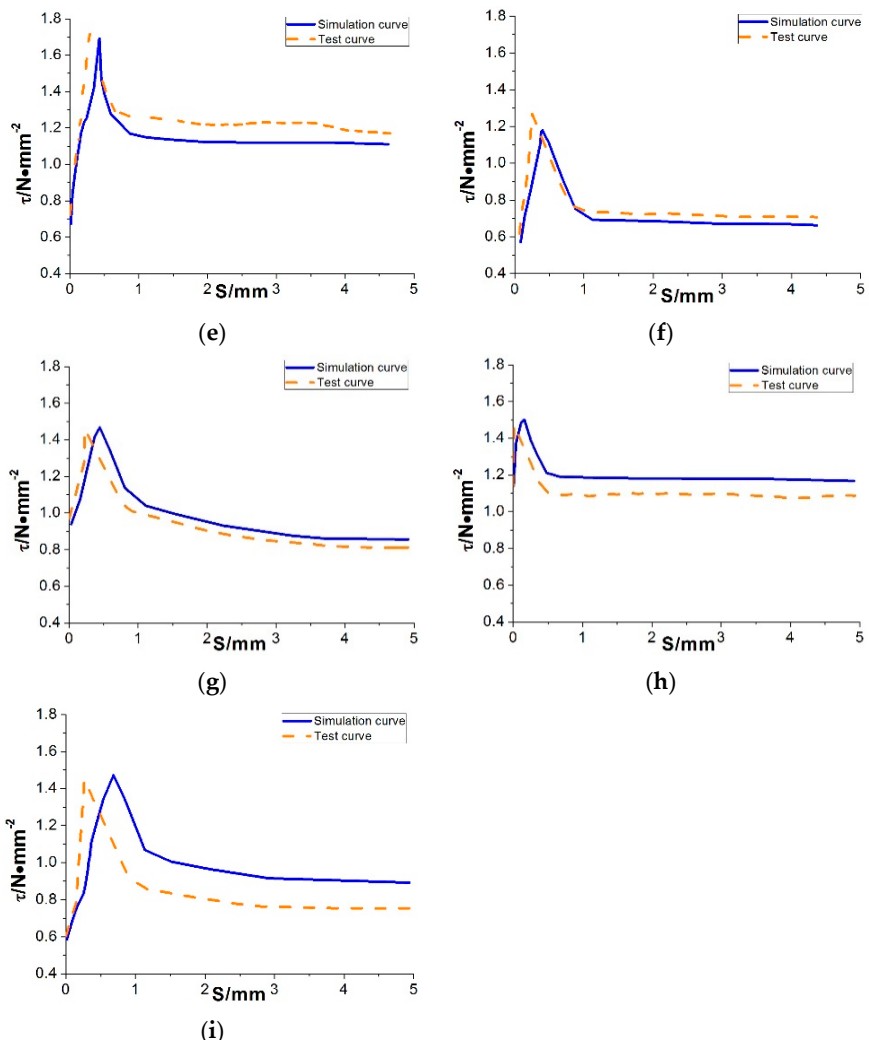

**Figure 17.** Simulation and test curves of τ-s: (**a**) SRRC-1; (**b**) SRRC-2; (**c**) SRRC-3; (**d**) SRRC-4; (**e**) SRRC-5; (**f**) SRRC-6; (**g**) SRRC-7; (**h**) SRRC-8; (**i**) SRRC-9.

It can be seen that the simulation results of initial bond strength, ultimate bond strength, and residual bond strength are basically equal with the test results. In the limit state, the slip value corresponding to the characteristic bond strength is larger than the test value, and the difference is within 0.15–0.65 mm. The slip value is close to the test value in the residual state, and the difference is within 0–0.15 mm.

In order to further verify the reliability of the simulation results, the experimental data in Chen et al. [24] and Yang et al. [40] were simulated by the numerical simulation method used in this study, and similar results were obtained.

### 4.2.2. Error Analysis

There are two reasons that the slip value corresponding to the characteristic bond strength is larger than the test value in the limited state:

(1) The bond stress–slip constitutive relationship between section steel and RAC is related to the embedded length of the section steel [32]. However, the constitutive relationship adopted in this study does not consider the influence of the position function, so there is an error in the slip value.

(2) From P–S curves of each specimen, it was found that the loading end and the free end slipped almost simultaneously, however, the free end began to slip when the loading end had reached the limited load. The finite element method (FEM) was also a factor that caused error.

## 5. Conclusions

In this study, nine push-out specimens were designed to study the bond behavior and the bond slip between section steel and RAC, and four factor effects of the concrete strength were investigated. Numerical analysis was conducted, and the simulation and test data were analyzed. The results are summarized as follows:

(1) The specimens were divided into splitting failure and bursting failure modes. The former is a typical failure mode, where the initial crack starts from the loading end and gradually extends to the free end. The latter is an atypical failure mode, where the initial crack in the middle of the flange side of the specimen gradually extends to both ends.

(2) The P–S curves were analyzed and classified into Type (I) and Type (II) according to the characteristic load. The initial load of the former is greater than the residual load value, and the latter is smaller than the residual load value. Type (II) occurs more easily due to increases of the cover thickness and the lateral stirrup ratio.

(3) The relationships between the characteristic bond strength and the concrete strength, the embedded length, the cover thickness, and the lateral stirrup ratio were analyzed. The characteristic bond strength increased with the increase of the concrete strength, the cover thickness, and the lateral stirrup ratio, and it decreased with the increase of the embedded length of the section steel.

(4) The FEM was used to simulate the specimens, and the simulation results were analyzed by comparing them with the experiment data. The analysis of the results showed that developed model is capable of representing the characteristic bond strength value between section steel and RAC with sufficient accuracy, and the main differences of bond slip between the simulation and the test results are the slippage at the limit state and the moment at which the free end starts to slip.

**Author Contributions:** Writing—review and editing, C.L., L.X. and H.L.; Resources, Z.Q., G.F., J.W., Z.L. and C.Z. All authors have read and agree to the published version of the manuscript.

**Funding:** This research was funded by [the National Natural Science Foundation of China] grant number [51878546], [the Innovative Talent Promotion Plan of Shaanxi Province] grant number [2018KJXX-056], [the Key Research and Development Program of Shaanxi Province] grant number [2018ZDCXL-SF-03-03-02], [the Key Laboratory Project of Shaanxi Province] grant number [XJKFJJ201904], [the Science and Technology Innovation Base of Shaanxi Province "Technology Innovation Service Platform for Solid Waste Resources and Energy Saving Wall Materials"] grant number [2017KTPT-19].

**Acknowledgments:** The authors would like to acknowledge the National Natural Science Foundation of China (Grant No. 51878546), the Innovative Talent Promotion Plan of Shaanxi Province (Grant No. 2018KJXX-056), the Key Research and Development Program of Shaanxi Province (Grant No. 2018ZDCXL-SF-03-03-02), the Key Laboratory Project of Shaanxi Province (Grant No. XJKFJJ201904), and the Science and Technology Innovation Base of Shaanxi Province "Technology Innovation Service Platform for Solid Waste Resources and Energy Saving Wall Materials" (2017KTPT-19) for financial support.

**Conflicts of Interest:** The authors declare no conflict of interest.

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
