# Peer review of "Numerical Study of Bond Slip between Section Steel and Recycled Aggregate Concrete with Full Replacement Ratio"

_applsci, doi:10.3390/app10030887_

Round 1
Reviewer 1 Report
As mentioned in the feedback above the English is rather poor with respect to some of the terminology used and also the way certain things are described.
As the experimental results have been used to determine the parameters of the springs used in the numerical simulation it is to be expected that the results would be comparable. The qualitative value of the results is however pertinent.
The fact that the fine aggregate and sand are "normal" rather than recycled does make me question the validity in call this wholly recycled aggregate concrete!
Reviewer 2 Report
The manuscript draft deals with the effects of the concrete strength, the embedded length, the cover thickness, and the lateral stirrup ratio on the bond behavior and the bond-slip of concrete prepared with recycled concrete (RAC) at full replacement ratio.
Based on the experimental data of bond behavior and the bond-slip between section steel and RAC measurement from nine push-out specimens, the p-s curves of the loading end were established. The numerical simulation using the Finite Element Method (FEM) was used.
The motivation of the present work is based on environmental and economical approach to reuse recycled aggregate concrete from construction demolition. The scientific background is the application of FEM to simulate the specimens
The main drawback of using RAC is the porosity of recycled aggregates and therefore, the pre-immersion was done before concrete preparation. Water in RAC pore can influence the course of hydration and moreover, RAC is composed of non-hydrated cement that can act as active fillers.
The work is well prepared and presented with excellent English. The contribution of this work is of great importance, as it deals with concrete using recycled aggregates at full replacement ratio.
It can be accepted.
Simple remark: unit of strength is MPa and not Mpa. Table 5
It could be useful to compare the results with ordinary concretes
